# Races *CYR34* and *Suwon11-1* of *Puccinia striiformis* f. sp. *tritici* Played an Important Role in Causing the Stripe Rust Epidemic in Winter Wheat in Yili, Xinjiang, China

**DOI:** 10.3390/jof9040436

**Published:** 2023-04-03

**Authors:** Li Chen, Muhammad Awais, Hong Yang, Yuyang Shen, Guangkuo Li, Haifeng Gao, Jinbiao Ma

**Affiliations:** 1State Key Laboratory of Desert and Oasis Ecology, Key Laboratory of Ecological Safety and Sustainable Development in Arid Lands, Xinjiang Institute of Ecology and Geography, Chinese Academy of Sciences, Urumqi 830000, China; 2Institute of Plant Protection, Xinjiang Academy of Agricultural Sciences/Key Laboratory of Integrated Pest Management on Crop in Northwestern Oasis, Ministry of Agriculture and Rural Affairs, Urumqi 830000, China; 3State Key Laboratory of Crop Stress Biology for Arid Areas, College of Plant Protection, Northwest A&F University, Xianyang 712100, China; 4College of Agriculture, Xinjiang Agricultural University/Key Laboratory of the Pest Monitoring and Safety Control of Crops and Forests, Urumqi 830000, China

**Keywords:** wheat stripe rust, races, Xinjiang, virulence, effective *Yr* genes

## Abstract

Wheat stripe rust caused by *Puccinia striiformis* f. sp. *tritici* is a destructive disease. Its pathogen frequently adapts to newly invaded regions and overcomes resistance in wheat cultivars. This disease is especially important in China due to its favorable conditions for the stripe rust epidemic and the recombination population structure of pathogens. Xinjiang is a vast epidemic region in China, but very limited research on this disease has been performed in this region. In this study, we identified 25 races from 129 isolates collected from winter wheat fields from five different regions (Nileke, Xinyuan, Gongliu, Huocheng, and Qapqal) of Yili, Xinjiang, using the Chinese set of 19 differential wheat lines. All isolates were virulent on the differentials Fulhad and Early Premium, but no isolates were virulent on *Yr5*. Among the 25 races, *Suwon11-1* was the most prevalent, followed by *CYR34*. Both races were found in four out of the five locations. It is important to continue monitoring stripe rust and its pathogen races in this region, as it forms a pathway between China and Central Asia. Collaborative research is essential for controlling stripe rust in this region, other regions in China, and neighboring countries.

## 1. Introduction

Plant genetic resistance is an effective approach to minimizing disease damage. The concept of plant resistance (R) began in the early 20th century [1]. Later, it was advanced by Harold Henry Flor’s groundbreaking research establishing the gene-for-gene model [2]. Hosts have multilayered defense mechanisms against pathogens [3]. On the other side, pathogens have evolved strategies to overcome the defense responses of their plant hosts [4]. Successful pathogens often evade detection by host R genes [5]. Thus, disease resistance conferred by a single R gene often lacks durability in the field because pathogens can evolve to evade recognition by mutating the corresponding avirulence (Avr) gene. Multiple R genes are often introduced simultaneously to improve durability and broaden the resistance spectrum [6,7].

*Puccinia striiformis* f. sp. *Tritici (Pst)* is one of the most destructive pathogens causing wheat stripe rust disease, and frequently mutates and overcomes resistance genes in major wheat cultivars [8,9,10]. The race identification of *Pst* in China was started in 1940 by Fang [11]. Chinese races were coded with *CYR* (Chinese yellow rust) and differential names series with a relative frequency of more than 10% [12]. Various cultivar replacement programs have taken place in China since the 1950s due to the repeated appearance of new virulent races [13]. At that time, *CYR1* broke down resistance in the wheat cultivar Bima 1 carrying the *Yr1* gene [14]. In the early 1970s, wheat cultivars and derivatives with the 1BL/1RS wheat-rye translocation chromosome, such as Lovrin 10, Lovrin 13, Predgornaia 2, Kavkaz, and Neuzucht, were used extensively in breeding programs in China because of their excellent resistance to stripe rust. However, now, more than 80% of commercially grown wheat cultivars (1B/1R and Fan6 derivatives) are susceptible to the predominant races *CYR31* and *CYR32* [15]. In recent years, the race *CYR34*, which is virulent to *Yr26*, *Yr24*, *Yr10* and other resistance genes, has become more prevalent [16], resulting in many wheat cultivars with *Yr26* becoming susceptible, such as Guinong 22 and its derived cultivars and Chuanmai 42 [17]. Resistance breeding faces many challenges due to the reduced availability of effective resistance genes. 

*Puccinia striiformis* f. sp. *tritici* urediniospores are dispersed by the wind and human activities. This species completes its life cycle on two host plants. Its asexual life cycle occurs in wheat, while sexual reproduction occurs on barberry plants [18]. To complete its life cycle, *P. striiformis* f. sp. *tritici* must avoid plant immune responses and absorb nutrients directly from the colonized plant tissue [19]. A previous study demonstrated the Himalayas region (Nepal, Pakistan and China) as a hot spot for sexual recombination and a centre of diversity. Meanwhile, in America, North Europe, East Africa and Australia, the fungus reproduces asexually [20]. China is the largest epidemic country worldwide for wheat stripe rust due to its large wheat production area and unique topography, providing favorable conditions for the pathogen’s survival and reproduction. In China, many Chinese *Berberis* species serve as alternate hosts of *Pst* [18,21,22,23]. Diverse agricultural systems, geography, climate, and pathogen life cycles in different regions create distinct disease epidemiological zones in China [24], which was recently divided into six main epidemic zones, including over-wintering (Chongqing, Central Shaanxi, South Shaanxi, Sichuan, Henan, North Hubei, and South Hubei), the eastern epidemic regions (Anhui, Zhejiang, Shandong, and Jiangsu), the Yun-Gui epidemic regions (Yunnan and Guizhou) and the Tibetan and Xinjiang epidemic regions. The southwestern and northwestern epidemic regions play a critical role in dispersing the pathogen in the central main wheat cultivation regions [25]. The disease is more prevalent in the winter-wheat-growing areas of the northwest, southwest, and north, and the spring-wheat-growing areas of northwest China [26]. Severe epidemics of this disease caused yield losses of 3.2 million metric tons (Mmt) in 1964, 1.8 Mmt in 1990, 1.3 Mmt in 2002, and 1.5 Mmt in 2017 in China, respectively [27]. 

Among different epidemic regions in China, the Xinjiang region is very important due to its location, geographically connecting the other regions of China with Central Asian countries, and it is a gateway for the spread of new races between Central Asian countries and China. As both winter wheat and spring wheat are cultivated in this region, the pathogen has living host plants for survival all year long. Additionally, many Chinese *Berberis* species have been identified as potential alternate hosts for *Pst* in the Xinjiang region [28], and stripe rust occurs in the neighboring countries, namely Tajikistan, Kyrgyzstan, Kazakhstan and Pakistan. The *Pst* populations in these countries are also phenotypically and genotypically diverse, especially in Pakistan [29]. Although recent studies suggest high divergence between the *Pst* populations in China and Pakistan [30], there is a potential risk of new race invasion from both countries. So, it is essential to have a regular stripe rust monitoring program in the Xinjian region to identify newly emerged races. Collaboration among scientists in China and Central Asian countries is needed for managing stripe rust.

The objectives of this study were to identify races and determine the virulence diversity of the *Pst* population in the winter wheat region of Yili, Xinjiang, China.

## 2. Materials and Methods

### 2.1. Wheat Stripe Rust Surveillance and Sample Collection

Wheat stripe rust surveillance was conducted in winter wheat fields in the Yili district, including Nileke, Gongliu, Xinyuan, Qapqal and Huocheng in the Xinjiang Autonomous Region of northwestern China from May to July 2021 following the protocol as described by Ali and Hodson [31]. The area of each of these counties ranges from 7850 to 70,650 square kilometers. Stripe rust samples were collected from five sites in each county. The sites were at least 10–15 km away from each other (Appendix A). Rust prevalence (percentage of plants infected), rust severity (percentage of leaf area infected), cultivar name, latitude, longitude, and altitude were recorded. Leaf samples with fresh single-stripe uredinia were collected from and packaged in a moisture-absorbent paper bag for drying for one to two days, and then the leaf samples were stored at 4 °C with a moisture-absorbing silica gel in an airtight bag.

### 2.2. Spore Multiplication and Preservation

For urediniospore multiplication, the leaf samples were first moisturized on a wet filter paper in a Petri dish at 10 °C for 10 h in darkness to induce fresh urediniospore production [32]. The 10–15-day-old seedlings of Mingxian 169, which is a susceptible wheat genotype, were inoculated with urediniospores from a single uredinium using a sterilized inoculating needle. Inoculated seedlings were sprayed with water mist and incubated in a dew chamber at 10 to 13 °C for 24 h in darkness. Then, the inoculated seedlings were maintained in a growth chamber with a day/night thermoperiod of 17/13 °C, a photoperiod of 16 h, relative humidity of 60%, and weekly watering. Fifteen days later, inoculated leaves began to sporulate. Urediniospores were collected in a test tube and used to inoculate new Mingxian 169 seedlings to produce an adequate amount of urediniospores. Urediniospores were collected in a clean glass tube and stored in a desiccator at 4 °C for a short period of time [33]. For long-term preservation, urediniospores were stored at −20 °C. Before use, urediniospores in the sealed glass tube were heat-shocked by submerging the tube in warm water (about 50 °C) for exactly 2 min. The urediniospore collection from a single leaf sample was treated as one isolate and used in virulence tests to identify the race.

### 2.3. Race Identification

The isolates were tested on the Chinese set of differentials including 19 wheat cultivars: 19 sets of Chinese-based differentials included Trigo-Eureka (*Yr6*), Fulhard (Unknown gene), Lutescens 128 (Unknown gene), Mentana (Unknown gene), Virgilio (*Yr Vir1, YrVir2*), Abbondanza (Unknown gene), Early Premium (Unknown gene), Funo (*YrA*,*+*), Danish 1 (*Yr3*), JubilejinaII (*YrJu1*, *YrJu2*, *YrJu3*, *YrJu4)*, Fengchan 3 (*Yr1*), Lovrin 13 (*Yr9*,*+*), Kangyin 655 (*Yr1*, *YrKy1*, *YrKy2)*, Suwon 11 (*YrSu*), Zhong 4 (Unknown gene), Lovrin 10 (*Yr9*), Hybrid 46 (*Yr3b*, *Yr4b)*, *Triticum spelta* var. Album (*Yr5*) and Guinong22 (*Yr10*, *Yr26)*, as described by Zhan et al. [16]. These differential lines carry single and multiple *Yr* genes. These differential lines were planted according to their numerical order with 5 varieties in each pot and the susceptible wheat genotype Mingxian 169 as a control. Fresh urediniospores and talcum powder were mixed at a 1:30 ratio in a 15 mL centrifuge tube and sealed with sterile gauze. Then, the mixture was gently shaken on seedlings of differentials with water mist. The plants were kept in the dew chamber and transferred to the growth chamber under the same conditions as described above. Each isolate’s reaction phenotypes on different differentials were recorded when the control genotype Mingxian 169 displayed full sporulation. If the virulence phenotypes on differential lines were unclear, the isolate was tested again for verification. Infection type was recorded after 18–20 days post inoculation, using the 0–9 scale, and classified as avirulent (IT 0–6) or virulent (IT 7–9).

Virulence phenotypes (previously identified races) were used to determine *Pst* races. Major virulence races were designated as CYR (Chinese Yellow Rust) races and some other races nominated with abbreviations of specific wheat differential genotypes, and new virulence patterns with a low frequency were temporary designated as P_1 series as described by Zhan et al. [16]. Frequencies and distribution, races and virulence factors on differential genotypes were analyzed. Cluster analyses were carried out using the Ward method [34] to assess the interrelationship of the location populations.

## 3. Results

### 3.1. Virulence Factors and Their Frequencies in Each Epidemic Region

A total of 129 isolates were obtained and tested on the Chinese set of 19 wheat differentials (Table 1). The virulence factors corresponding to the differential lines were coded as *Vr1, Vr2 … Vr19*. Virulence factor *Vr18* virulent on differential line 18 (*Triticum spelta* var. Album) with *Yr5* was not detected. In contrast, *Vr2* and *Vr7* on differential 2 (Fulhard) and differential 7 (Suwon 11 with YrSu) were detected in all isolates (Figure 1).

Across different locations, namely Nileke, Xinyuan, Gongliu, Qapqal, and Huocheng in the Xinjiang region, the maximum number of virulence factors was 18 in Xinyuan and Gongliu (Table 2). However, the virulence frequencies varied among areas. In Nileke, *Vr15* on differential 15 (Zhong 4) was not detected, whereas *Vr2*, *Vr3*, *Vr6*, *Vr7*, *Vr8*, *Vr9*, *Vr11*, and *Vr16* on differential lines Fulhad, Lutescens 128, Abbondanza, Early Premium, Funo, Danish 1, Fengchan 3 and Lovrin 10, respectively, were detected in all isolates. In Gongliu, *Vr2* and *Vr7* demonstrated 100% frequency. In Huocheng, 100% virulence frequency was detected in *Vr1*, *Vr2*, *Vr7*, and *Vr14*, while the other virulence factors were not detected. In the Qapqal and Xinyuan regions, *Vr2* and *Vr7* demonstrated a frequency of 100% (Table 3).

### 3.2. Race Groups through Cluster Analysis

From the 129 isolates, 21 previously reported races and 4 new races (P_10, P_18, P_20, P_23) were identified. The identified races were classified into four cluster groups (G1, G2, G3, and G4; Figure 2). G1 contains the races *Guinong22-14*, *CYR34*, and *Guinong22-13*, and these races showed a complex virulent pattern. These races were virulent on all differential lines except for *T. spelta* var. Album (*YR5*). G2 contains the races *Suowon11-1*, *Suwon11-2*, *Suwon11-10*, *Suwon11-8*, *Hy-6* and the newly identified races *P_20*, *P_10*, and *P_23*. The differential lines JubilejinaII, Lovrin 13, Kangyin 655, Zhong 4, Lovrin 10, *T. spelta* var. Album. and Guinong22 were resistant to the Group 2 races. G3 contains the races *CYR32*, *Hy-7*, *Hy-4*, *CYR33* and *Suwon11-4*. These races were virulent against all differential lines except for Zhong 4, *T. spelta* var. Album and Guinong22. G4 contains the races *CYR31*, *CYR30*, *CYR29*, *Lovrin13-2*, *CYR28*, *Suwon11-7* and *Suwon11-12*. The differential lines JubilejinaII, Kangyin 655 Zhong 4, *T. spelta* var. Album, Guinong22 were shown to be resistant to these races. These groups of races were not specific to one region, although race frequencies varied from region to region.

### 3.3. Races in Different Epidemic Regions

In Yili, the most prevailing race was *Suwon11-1*, with a frequency of 18.6%, followed by *CYR34* (14%). *CYR34* was more virulent compared to *Suwon11-1* (*Vr1*, *Vr2*, *Vr7*, *Vr14*), with 17 Vr virulence factors (*Vr1*, *Vr2*, *Vr3*, *Vr4*, *Vr5*, *Vr6*, *Vr7*, *Vr8*, *Vr9*, *Vr10*, *Vr11*, *Vr12*, *Vr13*, *Vr14*, *Vr16*, *Vr17*, and *Vr19*, Table 3).

Race diversity varies from region to region. The maximum race diversity was found in Xinyuan and Qapqal (Figure 3); some races were found only in specific locations, such as *P_20* and Suwon11-8, which were only detected in the Gongliu region. Races *CYR29*, *Hy7*, *Suwon11-4*, and *Suwon11-7* were specific to Qapqal, and *Hy-4* and *P_23* were specific to Xinyuan. The predominant race in Gongliu was *Suwon11-1* (22%), followed by *Suwon11-2* (19.5%). In Nileke, only three races, *CYR32*, *CYR34* and *Suwon11-12*, were detected. *Suwon11-1* was the prevailing race in Qapqal and Xinyuan, while *CYR34* was predominant in Xinyuan (Table 4).

## 4. Discussion

In the present study, we analyzed the virulence of stripe rust samples from the winter wheat region of Yili, Xinjiang in 2021. Xinjiang is the largest wheat producer in northwestern China, with an annual wheat growing area of about 1.13 million ha, accounting for 4% of the wheat area in China. It is classified as a separate wheat zone due to its unique geographic location and climatic conditions. This region is highly diverse and has different characteristics due to geographical subdivisions based on deserts, high mountains, and temperature fluctuations [36]. Furthermore, it is close to Central Asia. These spatial characteristics are highly influential for the dynamics of *P. striiformis* f. sp. *tritici* races [30]. There are approximately equal proportions of winter and spring wheat in Xinjiang. However, recent field surveys conducted in 2021 highlighted that winter wheat crops are severely affected by stripe rust. In this study, we found that *CYR34* and *Suwon11-1* were prevalent in Yili, with 14% and 18.6% frequencies, respectively. Previously, the race *CYR34*, which was virulent against Guinong 22 with *Yr26*, was absent in Xinjiang [16], but present in other epidemic regions of China [37]. This virulence against *Yr26* was also found in other continents, although it had low frequencies [20,38]. In recent years, reports of *CYR34* in the winter epidemic region of Xinjiang have presented an alarming situation for breeders. They need to develop resistant cultivars with high yield potential to replace the currently grown cultivars. Most recently, the breeding line 041133 was found to be highly resistant to *CYR34* [39].

Epidemiologically, the Xinjiang region is an independent region where *Pst* can survive in some areas all year long. The Xinjiang *Pst* population is genotypically different from the rest of the epidemic regions of China. However, many multilocus genotypes (MLG) have been detected in this area, suggesting that this region’s overall population was recombinant and could pose a potential threat to other regions [25]. Our study also detected 21 previously identified races and 4 new races. As the new races were observed at low frequencies, we assigned them temporary names in this study.

The deployment of resistance genes is the most economical and sustainable technique to control stripe rust [40,41]. So far, more than 80 resistance genes (*Yr1*–*Yr83*) have been discovered [12,42,43], and not all of them are widely used in wheat programs globally. *Yr26* has become ineffective against *CYR34* (V26) since 2011 [44]. Previous studies reported that 10% of Chinese winter wheat cultivars are resistant to *CYR34*. In our study, the *CYR* series races (*CYR28*–*CYR34*) were found in the Xinjiang region. CYR28 and CYR29 were first found in China in 1983–1985 [45] and belong to the Lv10/13 pathotype group. These races caused a massive epidemic in 1990 and affected 62% of the total planted area [12]. *CYR30*, which is virulent to Hybrid 46 (*Yr3b*, *Yr4b*, and *YrH46*), and *CYR31*, which is virulent against Hybrid 46 and Suwon 11, were first detected in 1991 in Sichuan and 1993 in Gansu, respectively, and became the predominant races in 1996 [46,47]. Our results revealed that in the overall population of Yili, Xinjiang, *Vr2* and *Vr7* had 100% frequency. In contrast, *Yr5* is still effective in this region. However, recently, a new race, TSA-6, was found to be virulent to *Yr5*. This race has a similar virulence pattern to the *CYR32* and *CYR34* races. It can evolve and generate new races [48,49]. Nowadays, wheat cultivars with the *Yr9* gene are susceptible to stripe rust worldwide. The breakdown of *Yr9* has been reported since the late 1980s due to the widespread use of this gene in wheat production worldwide, resulting in major epidemics of stripe rust [50,51]. In China, the first cases of the *Yr9*-virulent races *Lv10* and *Lv13* were detected in Longnan in Gansu province [52,53]. Our study also reported virulence against *Yr9* in the winter wheat crop in the Xinjiang region. As the genetic bases of resistance in most Chinese differentials used in the present study are not fully understood, it is difficult to compare Chinese *Pst* races with those in other countries.

There are several reasons for the variation in virulence in Xinjiang regions. One could be the presence of *Berberis* species, identified as potential alternate hosts for *Pst* in the Xinjiang region [28]. The long-distance dispersal of the pathogen by the wind may also be responsible for its racial diversity. The cultivars grown in this region, which are different from those grown in other regions, might have selected and maintained unique races. The growing of both winter and spring wheat crops in this region may also increase the pathogen’s variation. More studies are needed to characterize *Pst* populations on spring wheat.

## 5. Conclusions

In this study, race identification was conducted in the Yili region of Xinjiang, and 25 races were identified. *CYR34* and *Suwon11-1* were the top two predominant races. The wheat differentials Fulhard and Early Premium lines were susceptible, whereas *Yr5* in the differential *T. spelta* var. Album was still effective. It is essential to understand the population structure of *Pst* in Xinjiang and its genetic relationships with the populations in other regions, especially in neighboring Central Asian countries. Such information should provide the basis for developing effective strategies for the control of the disease.

## Figures and Tables

**Figure 1 jof-09-00436-f001:**
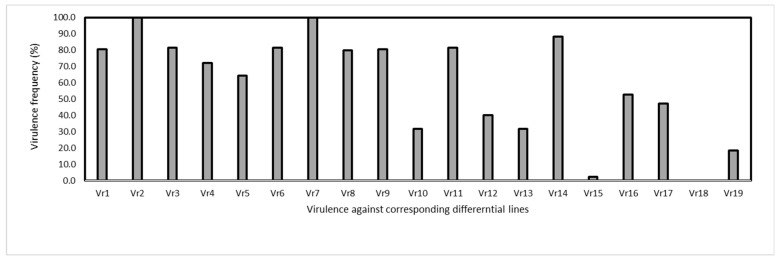
Frequencies of virulence factors detected in the *Puccinia striiformis* f. sp. *tritici* isolates collected from winter wheat in Yili, Xinjiang, China in 2021.

**Figure 2 jof-09-00436-f002:**
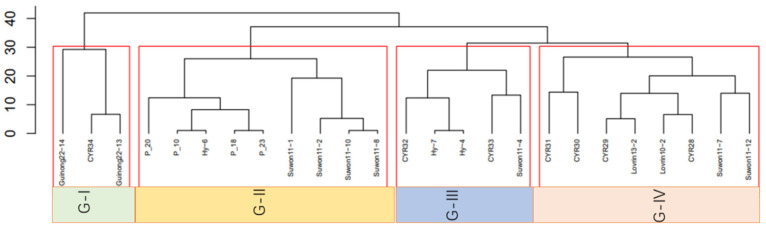
Clustering of 25 races detected from 129 isolates of *Puccinia striiformis* f. sp. *tritici* collected from different locations in Xinjiang province of China in 2021.

**Figure 3 jof-09-00436-f003:**
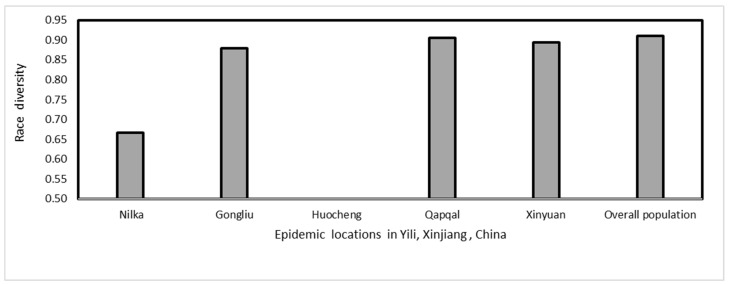
Race diversity in *Puccinia striiformis* f. sp. *tritici* isolates collected from Yili, Xinjiang, China, in 2021.

**Table 1 jof-09-00436-t001:** Chinese differential lines used to identify races of *Puccinia striiformis* f. sp. *tritici* isolates collected from Yili, Xinjiang, China, in 2021.

Differential Order	Virulence	Differential Lines	Yr Gene ^a^
1	*Vr1*	Trigo-Eureka	*Yr6*
2	*Vr2*	Fulhard	Unknown
3	*Vr3*	Lutescens 128	Unknown
4	*Vr4*	Mentana	Unknown
5	*Vr5*	Virgilio	*YrVir1*, *YrVir2*
6	*Vr6*	Abbondanza	Unknown
7	*Vr7*	Early Premium	Unknown
8	*Vr8*	Funo	*YrA*,*+*
9	*Vr9*	Danish 1	*Yr3*
10	*Vr10*	JubilejinaII	*YrJu1*, *YrJu2*, *YrJu3*, *YrJu4*
11	*Vr11*	Fengchan 3	*Yr1*
12	*Vr12*	Lovrin 13	*Yr9*,*+*
13	*Vr13*	Kangyin 655	*Yr1*, *YrKy1*, *YrKy2*
14	*Vr14*	Suwon 11	*YrSu*
15	*Vr15*	Zhong 4	Unknown
16	*Vr16*	Lovrin 10	Yr9
17	*Vr17*	Hybrid 46	*Yr3b*, *Yr4b*
18	*Vr18*	*Triticum spelta* var. *Album*	*Yr5*
19	*Vr19*	Guinong22	*Yr10* , *Yr26*

^a^ The *Yr* genes in differential lines refer to Wan et al. [27], Chen et al. [35] and Zhan et al. [16].

**Table 2 jof-09-00436-t002:** Frequencies (%) of virulence factors in *Puccinia striiformis* f. sp. *tritici* isolates collected from different locations in Xinjiang province of China in 2021.

Virulence	Nileke	Gongliu	Huocheng	Qapqal	Xinyuan	Overall Population
*Vr1*	66.7	82.9	100.0	70.4	84.2	80.6
*Vr2*	100.0	100.0	100.0	100.0	100.0	100.0
*Vr3*	100.0	78.0	0.0	77.8	86.0	81.4
*Vr4*	66.7	70.7	0.0	63.0	78.9	72.1
*Vr5*	66.7	58.5	0.0	51.9	75.4	64.3
*Vr6*	100.0	78.0	0.0	77.8	86.0	81.4
*Vr7*	100.0	100.0	100.0	100.0	100.0	100.0
*Vr8*	100.0	75.6	0.0	77.8	84.2	79.8
*Vr9*	100.0	75.6	0.0	77.8	86.0	80.6
*Vr10*	66.7	19.5	0.0	25.9	42.1	31.8
*Vr11*	100.0	78.0	0.0	77.8	86.0	81.4
*Vr12*	66.7	26.8	0.0	33.3	52.6	40.3
*Vr13*	66.7	19.5	0.0	25.9	42.1	31.8
*Vr14*	100.0	92.7	100.0	81.5	87.7	88.4
*Vr15*	0.0	2.4	0.0	0.0	3.5	2.3
*Vr16*	100.0	34.1	0.0	59.3	61.4	52.7
*Vr17*	66.7	34.1	0.0	40.7	59.6	47.3
*Vr18*	0.0	0.0	0.0	0.0	0.0	0.0
*Vr19*	33.3	7.3	0.0	14.8	28.1	18.6
Sample size	3	41	1	27	57	129

**Table 3 jof-09-00436-t003:** Number of different pathotypes, virulences and their diversity detected in the *Puccinia striiformis* f. sp. *tritici* isolates collected from different locations in Xinjiang province of China in 2021.

Virulence	Sample Size	Number of Races Detected	Race Diversity	Frequency (%) of the Most Frequent Race	Number of Virulences Detected	Frequency (%) of the Most Frequent Virulence
Nileke	3	3	0.67	33.33	17	100
Gongliu	41	16	0.88	21.95	18	100
Huocheng	1	1	0.00	100.00	4	100
Qapqal	27	16	0.91	22.22	17	100
Xinyuan	57	17	0.89	22.81	18	100
Overall population	129	25	0.91	18.60	18	100

**Table 4 jof-09-00436-t004:** Frequencies (%) of different races detected from *Puccinia striiformis* f. sp. *tritici* isolates collected from different locations in Xinjiang province of China in 2021.

Race	Virulence Profile ^a^	Nileke	Gongliu	Huocheng	Qapqal	Xinyuan	Overall Population
*CYR28*	1,2,3,4,5,6,7,8,9,-,11,-,-,-,-,16,-,-,-	-	2.4	-	7.4	3.5	3.9
*CYR29*	1,2,3,4,5,6,7,8,9,-,11,12,-,-,-,16,-,-,-	-	-	-	3.7	-	0.8
*CYR30*	1,2,3,4,5,6,7,8,9,-,11,12,-,-,-,16,17,-,-	-	2.4	-	-	7.0	3.9
*CYR31*	1,2,3,4,5,6,7,8,9,-,11,12,-,14,-,16,17,-,-	-	4.9	-	7.4	5.3	5.4
*CYR32*	1,2,3,4,5,6,7,8,9,10,11,12,13,14,-,16,17,-,-	33.3	12.2	-	-	8.8	8.5
*CYR33*	-,2,3,4,5,6,7,8,9,10,11,12,13,14,-,16,-,-,-	-	-	-	-	1.8	0.8
*CYR34*	1,2,3,4,5,6,7,8,9,10,11,12,13,14,-,16,17,-,19	33.3	4.9	-	7.4	22.8	14.0
*Guinong22-13*	-,2,3,-,-,6,7,8,9,10,11,12,13,14,-,16,17,-,19	-	-	-	7.4	1.8	2.3
*Guinong22-14*	1,2,3,4,5,6,7,8,9,10,11,12,13,14,15,-,-,-,19	-	2.4	-	-	3.5	2.3
*Hy-4*	1,2,3,4,5,6,7,8,9,10,11,-,13,14,-,16,17,-,-	-	-	-	-	3.5	1.6
*Hy-6*	1,2,3,4,5,6,7,8,9,-,11,-,-,14,-,-,17,-,-	-	2.4	-	7.4	5.3	4.7
*Hy-7*	-,2,3,4,5,6,7,8,9,10,11,-,13,14,-,16,17,-,-	-	-	-	7.4	-	1.6
*Lovrin10-2*	1,2,3,-,-,6,7,8,9,-,11,-,-,-,-,16,-,-,-	-	2.4	-	3.7	-	1.6
*Lovrin13-2*	1,2,3,4,-,6,7,8,9,-,11,12,-,-,-,16,-,-,-	-	-	-	3.7	1.8	1.6
*P_10*	-,2,3,4,5,6,7,8,9,-,11,-,-,14,-,-,17,-,-	-	2.4	-	3.7	3.5	3.1
*P_18*	1,2,3,4,5,6,7,-,9,-,11,-,-,14,-,-,17,-,-	-	2.4	-	-	-	0.8
*P_20*	-,2,3,4,5,6,7,8,-,-,11,-,-,14,-,-,17,-,-	-	2.4	-	-	-	0.8
*P_23*	-,2,3,4,5,6,7,-,9,-,11,-,-,14,-,-,17,-,-	-	-	-	-	1.8	0.8
*Suwon11-1*	1,2,-,-,-,-,7,-,-,-,-,-,-,14,-,-,-,-,-	-	22.0	100	22.2	14.0	18.6
*Suwon11-10*	-,2,3,4,-,6,7,8,9,-,11,-,-,14,-,-,-,-,-	-	7.3	-	3.7	1.8	3.9
*Suwon11-12*	-,2,3,-,-,6,7,8,9,-,11,-,-,14,-,16,-,-,-	33.3	4.9	-	3.7	5.3	5.4
*Suwon11-2*	1,2,3,4,5,6,7,8,9,-,11,-,-,14,-,-,-,-,-	-	19.5	-	3.7	8.8	10.9
*Suwon11-4*	-,2,3,4,-,6,7,8,9,10,11,-,13,14,-,16,-,-,-	-	-	-	3.7	-	0.8
*Suwon11-7*	1,2,3,4,5,6,7,8,9,-,11,12,-,14,-,16,-,-,-	-	-	-	3.7	-	0.8
*Suwon11-8*	1,2,3,4,-,6,7,8,9,-,11,-,-,14,-,-,-,-,-	-	4.9	-	-	-	1.6
Sample size		3	41	1	27	57	129

^a^ Virulence profile showed race virulence pattern against differential lines: 1 = Trigo Eureka, 2 = Fulhard, 3 = Lutescens 128, 4 = Mentana, 5 = Virgilio, 6 = Abbondanza, 7 = Early Premium, 8 = Funo, C9 = Danish 1, 10 = Jubilejina 2, 11 = Fengchan 3, 12 = Lovrin 13, 13 = Kangyin 655, 14 = Suwon 11, 15 = Zhong 4, 16 = Lovrin 10, 17 = Hybrid 46, 18 = *Triticum spelta* var. Album, and 19 = Guinong 22.

## Data Availability

Original data presented in this paper are available within paper. If further detail needed, contact corresponding authors.

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
