# Peer review of "Races CYR34 and Suwon11-1 of Puccinia striiformis f. sp. tritici Played an Important Role in Causing the Stripe Rust Epidemic in Winter Wheat in Yili, Xinjiang, China"

_jof, 2023, doi:10.3390/jof9040436_

Round 1
Reviewer 1 Report
Comments to “Race CYR34 and Suwon11-1 of Puccinia striiformis f.sp. tritici played an important role causing the Stripe rust epidemic on winter wheat in Yili, Xinjiang, China, during cropping season 2021”
The manuscript describes the identification of 25 races/pathotypes of Puccinia striiformis f.sp. tritici in the Xinjiang region of China, along with virulence profiles, race diversity and relative frequency, providing also information on which resistance genes are still effective and which other are breakdown. The topic is of main importance, also because, as the authors point out “Xinjiang regions have great importance due to its strategic location, which links China with Central Asian countries”.
Experimental are essentially well conducted, even if, in my point of view, molecular approaches would have completed information. The argument is complex because Pst isolate- host wheat variety interactions are complex and dynamics, for this reason exposition should be clear and rigorous, that is not always the case.
For example the terms "races" and "pathotypes" should be clearly explained as in Chen W. et al 2014: “The major virulence patterns have been formally designated ‘CYR’ races (Chinese Yellow Rust) with sequential numbers based on their chronological identification. Virulence patterns with low frequencies and limited distribution were temporarily nominated as ‘pathotypes’ using the abbreviations of specific wheat differential genotypes.”
Also the “race identification test” is not clear.
English should be revised, I gave some suggestions but specialist revision is suggested. Please check punctuation!
A supplementary table is mentioned but I didn’t find it in the non-published material folder, that result empty. Different References are missing!!
These and other corrections and suggestions are reported in the revised manuscript file.
Overall I recommend publication after requested revisions.

Author Response
We are highly thankful to the reviewer for his valuable time on this article and suggestions —the answer to the reviewer's comments are given below.
Comments to “Race CYR34 and Suwon11-1 of Puccinia striiformis f.sp. tritici played an important role causing the Stripe rust epidemic on winter wheat in Yili, Xinjiang, China, during cropping season 2021”
The manuscript describes the identification of 25 races/pathotypes of Puccinia striiformis f.sp. tritici in the Xinjiang region of China, along with virulence profiles, race diversity and relative frequency, providing also information on which resistance genes are still effective and which other are breakdown. The topic is of main importance, also because, as the authors point out “Xinjiang regions have great importance due to its strategic location, which links China with Central Asian countries”.
Experimental are essentially well conducted, even if, in my point of view, molecular approaches would have completed information. The argument is complex because Pst isolate- host wheat variety interactions are complex and dynamics, for this reason exposition should be clear and rigorous, that is not always the case.
For example the terms "races" and "pathotypes" should be clearly explained as in Chen W. et al 2014: “The major virulence patterns have been formally designated ‘CYR’ races (Chinese Yellow Rust) with sequential numbers based on their chronological identification. Virulence patterns with low frequencies and limited distribution were temporarily nominated as ‘pathotypes’ using the abbreviations of specific wheat differential genotypes.”
Also the “race identification test” is not clear.
Answer: Thanks for valuable feedback. The races were named according to previously identified race virulence patterns explained by Chen et al. 2009 and zhan et al. 2016 (Race identification method added in Line 144-147). Pathotype with new virulence patterns with the low frequency were named as temporary race as previously described by Zhan et al. 2016. The word pathotype was replaced throughout the paper, as also suggested by another reviewer.
English should be revised, I gave some suggestions but specialist revision is suggested. Please check punctuation!
Answer: English is revised as suggested.
A supplementary table is mentioned but I didn’t find it in the non-published material folder, that result empty. Different References are missing!!
Answer: supplement table is added to the attached email. I will request the editor to add it to the supporting material folder. Missing references were added to the manuscript and revised according to the journal format
These and other corrections and suggestions are reported in the revised manuscript file.
Overall I recommend publication after requested revisions.
Answer: Again thank you to recommend this paper for publication. Other Suggestions are revised inside manuscript with track mark changes and directly answer to comments accordingly in manuscript.
Reviewer 2 Report
The study was conducted quite well, and the results are significant. The manuscript was not written well due to poor English writing. I marked intensive changes and comments directly in the manuscript, which should improve the manuscript. After the authors make all of the changes, it can be accepted for publication.

Author Response
Comments and Suggestions for Authors
The study was conducted quite well, and the results are significant. The manuscript was not written well due to poor English writing. I marked intensive changes and comments directly in the manuscript, which should improve the manuscript. After the authors make all of the changes, it can be accepted for publication.
Answer: We are highly thankful to you to spent your time on this manuscript and specially your suggestion to revise manuscript style and English which improved overall manuscript quality. All of your suggestion and recommendation was revised as suggested inside manuscript mark with track changes. Also missing reference is inserted as suggested and reference style reformat according to journal criteria.